# Placental Origins of Preeclampsia: Insights from Multi-Omic Studies

**DOI:** 10.3390/ijms25179343

**Published:** 2024-08-28

**Authors:** Chang Cao, Richa Saxena, Kathryn J. Gray

**Affiliations:** 1Center for Genomic Medicine and Department of Anesthesia, Critical Care and Pain Medicine, Massachusetts General Hospital, Harvard Medical School, Boston, MA 02114, USA; 2Division of Maternal Fetal Medicine, Department of Obstetrics and Gynecology, University of Washington School of Medicine, Seattle, WA 98195, USA

**Keywords:** placenta, preeclampsia, single-cell RNA sequencing, proteomics, genome-wide association studies, multi-omics

## Abstract

Preeclampsia (PE) is a major cause of maternal and neonatal morbidity and mortality worldwide, with the placenta playing a central role in disease pathophysiology. This review synthesizes recent advancements in understanding the molecular mechanisms underlying PE, focusing on placental genes, proteins, and genetic variants identified through multi-omic approaches. Transcriptomic studies in bulk placental tissue have identified many dysregulated genes in the PE placenta, including the PE signature gene, Fms-like tyrosine kinase 1 (*FLT1*). Emerging single-cell level transcriptomic data have revealed key cell types and molecular signatures implicated in placental dysfunction and PE. However, the considerable variability among studies underscores the need for standardized methodologies and larger sample sizes to enhance the reproducibility of results. Proteomic profiling of PE placentas has identified numerous PE-associated proteins, offering insights into potential biomarkers and pathways implicated in PE pathogenesis. Despite significant progress, challenges such as inconsistencies in study findings and lack of validation persist. Recent fetal genome-wide association studies have identified multiple genetic loci associated with PE, with ongoing efforts to elucidate their impact on placental gene expression and function. Future directions include the integration of multi-omic data, validation of findings in diverse PE populations and clinical subtypes, and the development of analytical approaches and experimental models to study the complex interplay of placental and maternal factors in PE etiology. These insights hold promise for improving risk prediction, diagnosis, and management of PE, ultimately reducing its burden on maternal and neonatal health.

## 1. Introduction

Preeclampsia (PE) is a hypertensive disorder affecting 3–7% of all pregnancies in the United States [1] and 1–10% globally [2]. PE is a leading cause of maternal and neonatal mortality worldwide, resulting in the deaths of over 70,000 pregnant individuals and 500,000 fetuses/neonates each year [2]. PE leads to end-organ dysfunction across multiple systems and is associated with increased long-term cardiometabolic disease risks in both the pregnant person and the offspring [3].

While the causal pathways leading to PE are not fully characterized, the placenta is known to be central to PE pathophysiology [3], with delivery of the placenta being the only cure [4]. Alterations in placental function, including inadequate vascular remodeling and aberrant production of anti-angiogenic and pro-inflammatory factors (e.g., soluble form of fms-like tyrosine kinase 1 (sFlt1), placental growth factor (PlGF), soluble form of endoglin), are recognized as key contributors to the initiation and systemic manifestations of PE [2,3]. The identification of placental-origin PE biomarkers has improved PE prediction in clinical settings. For instance, measuring serum PIGF [5] and the sFlt1/PIGF ratio [6] enhances the diagnosis and prediction of PE in pregnant individuals being evaluated for PE. Prediction models that incorporate serum PIGF with clinical risk factors, such as the model developed by the Fetal Medicine Foundation in the United Kingdom, have shown high predictive value for PE across racially diverse populations worldwide [7,8]. These models are also increasingly adopted by regional and international guidelines to guide PE management strategies, including aspirin prophylaxis [9]. As many at-risk pregnancies are still missed by current PE prediction models, ongoing work in improving prediction remains critical.

Advances in multi-omic technologies have accelerated the discovery of genes and proteins implicated in PE (reviewed by Benny et al. [10]). In particular, transcriptomic studies of bulk placental tissue have identified many PE-associated genes, transcription factors, and metabolic pathways [10,11,12]. However, meta-analyses consistently show a lack of consensus among studies [11,12], and the specific functions of these PE-associated genes and pathways in the placenta remain poorly characterized and under-reviewed. Recent single-cell RNA sequencing (scRNA-seq) studies of the PE placenta have revealed distinct molecular signatures in early-onset PE (EOPE) (delivery < 34 weeks gestation) and late-onset PE (LOPE) (delivery ≥ 34 weeks gestation) and identified critical placental cell types and molecular pathways involved in PE pathology [13]. Furthermore, a comprehensive summary of all PE-associated placental proteins identified with proteomic analyses over the past two decades is now available for close review [14]. Additionally, large-scale genome-wide association studies (GWASs) of PE are beginning to identify risk variants in the maternal and fetal genomes, including those associated with placentally enriched genes *FLT1* and Wnt family member 3A (*WNT3A*), which likely contribute to PE risk due to their effects on placental gene expression [15,16,17].

In this review, we highlight the pivotal role of the placenta in PE. We summarize recent findings from scRNA-seq studies of PE placentas, identified through a PubMed search using the keywords “placenta”, “preeclampsia”, and “single-cell RNA sequencing”. We examine the literature on placental expression and function of proteins consistently altered in the PE placental proteome, exploring their mechanistic links to PE pathophysiology. Additionally, we discuss the implications of maternal and fetal GWAS findings for identifying placental cells and genes that confer increased PE risk. Lastly, we outline research gaps and propose how novel analytic approaches and multi-omic data can enhance our understanding of PE.

## 2. Transcriptomic Changes in the PE Placenta

Over the past two decades, transcriptomic studies have identified numerous differentially expressed genes in placentas from pregnancies with PE, including the widely recognized anti-angiogenic gene, *FLT1* [18]. Differentially expressed genes in the PE placenta are involved in pathways that affect energy utilization, signal transduction, and innate immunity [11]. Findings from bulk transcriptomic studies are reviewed elsewhere [10,11,12].

Despite many available datasets, meta-analyses consistently show poor overlap of PE-associated transcriptomic signatures across studies [11,12,19]. This inconsistency likely arises from variations in PE subtype, study population, gestational age at delivery, sample size, placental cell composition, and experimental methodology. Deconvolution of bulk placental transcriptomes has revealed that PE placentas have a higher proportion of extravillous trophoblast (EVT) cells [20,21] and that differences in placental cell composition account for substantial variation between PE and control transcriptomes [20]. These findings indicate that altered cell composition is a key feature of PE placentas and should be considered when interpreting bulk gene expression measures. Given the cellular complexity and spatial heterogeneity within the placenta, it is critical to study cell-specific expression to identify the cell types and pathways that drive PE pathophysiology.

scRNA-seq has provided powerful insights into the cellular dynamics and biological processes at the maternal–fetal interface [22,23]. The PE research community has embraced single-cell-based approaches, generating a diverse array of scRNA-seq data since 2017 [13,24,25,26,27,28] (Table 1). Despite variations in study design, most placental scRNA-seq investigations have captured the same set of cell types, including syncytiotrophoblast (SCT), EVT, villous cytotrophoblasts (VCTs), endothelial cells, and macrophages. Notably, EVT and macrophages exhibit different abundances in PE placentas compared to controls, consistent with findings from bulk transcriptome deconvolution studies [20]. Furthermore, EVT is consistently identified as the cell type with altered immune response and invasion and increased cell death in the PE placenta.

Like transcriptomic studies of bulk placental tissue, scRNA-seq studies show significant variability in the PE-associated genes and pathways identified. Several factors contribute to this variability, including small sample sizes, PE case heterogeneity, low cell numbers, and RNA sequencing depth, which hinder the detection of rare cell types and low-expression genes. Furthermore, the specificity of many marker genes for cell type inference has not been robustly evaluated in the placenta, and there is a lack of consensus on markers for annotating placental cell types [29]. Consequently, studies often employ different marker genes for the same cell type, further contributing to observed interstudy variation.

Another primary source of variability is the use of term placentas as controls for PE placentas, which are frequently from preterm deliveries. This leads to transcriptomic differences that reflect both disease status and gestational age of assessment. Using gestational age-matched controls, Admati et al. [13] demonstrated that EOPE placentas exhibit substantial transcriptomic changes across all major cell types, whereas LOPE placentas show minimal alterations (Table 1). The striking differences between the EOPE and LOPE single-cell transcriptome are also observed in bulk placental tissue [30]. Collectively, these studies provide compelling molecular evidence for the distinct placental pathophysiology underlying EOPE versus LOPE, emphasizing the importance of using well-phenotyped PE cases and controls to generate clinically relevant findings. However, it is important to note that the control placentas for EOPE in the study by Admati et al. [13] were from spontaneous preterm deliveries, which may exhibit heightened inflammation and other metabolic abnormalities compared to normal gestation. Therefore, while gestational age-matched placentas help reduce variability, they may not fully capture the transcriptomic differences between PE and healthy placentas. More data is needed to establish the normal expression profiles of placentas in early gestation.

Similar investigations into other PE subtypes, including HELLP (hemolysis, elevated liver enzymes, low platelets) syndrome and related conditions such as acute fatty liver of pregnancy, are needed to identify their molecular hallmarks and unique pathophysiology. Additionally, exploring whether the fetal/placental sex affects the PE placental transcriptome could provide insights into sex-specific effects on disease [31].

While most transcriptomic analyses of PE placentas have focused on protein-coding mRNAs, there is growing interest in understanding the role of small RNA species in both PE etiology and prediction. A recent high-coverage small RNA-seq study of 164 PE and healthy placentas uncovered many differentially expressed non-coding RNAs, including 43 small RNAs, 12 miRNAs, and 2 circular RNAs [32]. Additionally, placental exosomes in maternal plasma from PE pregnancies show elevated levels of miRNAs implicated in cell proliferation and apoptosis [33]. More studies are needed to replicate these findings and determine the functional relevance of circulating RNAs of placental origin in PE development. Small RNA-seq of PE placentas at the single-cell level may help uncover cell type-specific small RNA species and miRNA–mRNA interactions in placental dysfunction associated with PE.

More studies like those by Tsang et al. [28], Moufarrej et al. [34], and Rasmussen et al. [35] are needed to characterize the temporal changes in placentally-derived RNA in maternal plasma across pregnancy and identify circulating biomarkers that reflect placental function and pathology. Importantly, reevaluating previously reported circulating PE signatures using improved placental cell markers and analytical techniques, as demonstrated by Vorperian et al. [36], is necessary to validate the biological relevance, reproducibility, and prognostic utility of potential PE biomarkers [37].

## 3. Proteomic Changes in the PE Placenta

Since 2007, proteomic studies have identified over 900 PE-associated placental proteins. However, there is considerable variability between studies and a lack of validation of the identified proteins [10,14].

A recent meta-analysis of 23 proteomic studies of PE placentas [14] identified eleven proteins that show consistent changes in at least three studies (Table 2). Among these, eight proteins, including FLT1 and pappalysin 2 (PAPPA2), are upregulated, while three proteins, actin gamma 1 (ACTG1), fibrinogen beta chain (FGB), and albumin (ALB), are downregulated in PE placentas.

Three of the eleven PE-associated placental proteins, hemoglobin subunit zeta (HBZ), FGB, and ALB, are blood/plasma proteins. HBZ is the alpha-like globin component of fetal hemoglobin in the early embryo [66]. As gestation progresses, HBZ production ceases and is replaced by alpha globin. Thus, HBZ in the placental proteome most likely derives from fetal red blood cells. Elevated HBZ in the PE placenta likely indicates an earlier gestational age of the placental sample compared to the control, rather than being causal for PE.

Similarly, ALB, the most abundant plasma protein, may not directly contribute to PE pathology, although there is evidence for its involvement in regulating placental endothelial function [67]. Given that many hormones and metabolites are bound by albumin, lower levels of ALB in the PE placenta may increase their bioavailability within the placenta. Additionally, up to 85% of the placental proteome consists of blood proteins [68]. Thus, reduced ALB in PE placentas likely mirrors decreased maternal plasma ALB levels [69,70] rather than indicating direct participation in PE development.

FGB, a component of fibrinogen responsible for forming fibrin-based clots during vascular injury, is another plasma protein altered in the PE placenta. Plasma fibrinogen levels typically rise progressively in normal pregnancy [71]. Data on plasma fibrinogen in PE pregnancies are limited and show inconsistent changes compared to healthy controls [72]. It is unclear why the PE placental proteome has lower FGB. In healthy placentas, FGB is present in fetal blood vessels [65], and there is evidence suggesting that EOPE placentas have fewer fibrinogen-positive fetal endothelial cells than controls [64]. Further studies are needed to investigate the role of fibrinogen in vascular homeostasis within the placenta during pregnancy and its connection to the fibrin deposits frequently observed in PE placentas [73].

Except for the three proteins of blood origin, the remaining eight PE-associated placental proteins are expressed by placental trophoblasts, particularly SCT and EVT. Among these, the associations between soluble FLT1 (sFLT1) and PAPPA2 with PE are well characterized and validated (Table 2).

sFLT1, the soluble form of FLT1, is released into the maternal circulation from the placenta during gestation, with higher levels observed in PE patients [2,18]. sFLT1 has been shown to play a causal role in the endothelial dysfunction noted across multiple organs in PE, including blood vessels, liver, and kidneys [2]. FLT1 is a member of the vascular endothelial growth factor (VEGF) receptor family and plays a critical role in angiogenesis [74]. Notably, *FLT1* mRNA expression is the highest in the placenta compared to other human tissues [75]. In the first-trimester placenta, FLT1 is localized to SCT and EVT [57], and it is additionally found in VCT and fetal endothelium in the third-trimester placenta [75]. Placental FLT1 expression is elevated in PE [53,54] and positively correlates with maternal plasma sFLT1 levels [54]. Immunohistochemistry (IHC) studies demonstrate that FLT1 elevation in PE placentas is most pronounced in the SCT [19,53,55,56,57]. These observations support the current understanding that increased sFLT1 in PE maternal circulation primarily originates from the placenta [4,18,19].

PAPPA2 is a protease for insulin-like growth factor binding proteins (IGFBP) 3 and 5, potentially enhancing growth factor bioavailability [76]. Plasma levels of PAPPA2 [77,78] and its homolog metalloproteinase, pappalysin 1 (PAPPA) [77,79] have shown predictive values for PE risk. Like FLT1, PAPPA2 is highly enriched in the placenta [75] and is localized to SCT and EVT [65]. Both EOPE and LOPE placentas exhibit increased PAPPA2 expression compared to controls [59,60,61,62]. IHC has localized this PE-associated increase to SCT, with some data also showing enhanced staining in EVT [59], while others reporting no such difference [61].

Investigations into the physiological role of placental PAPPA2 suggest it is unlikely to be a causal factor of PE. The deletion of *Pappa2* in mice does not affect fetal growth [80], and alterations in placental PAPPA2 have no effect on fetal weight or placental mass [81]. Interestingly, trophoblast PAPPA2 can be induced by two well-known molecular hallmarks of PE: hypoxia and pro-inflammatory cytokines [82]. Similarly, PAPPA2 production in EVT is stimulated by uterine natural killer cell cytokines and is associated with reduced IGFBP1-3 [83]. These findings suggest that elevated PAPPA2 in PE placentas likely reflects a response to the immune milieu of PE rather than being a direct cause of its pathology [77,82].

Compared to FLT1 and PAPPA2, the relationships between PE and the other proteomics-identified placental proteins are poorly understood. Among these proteins, PE-associated changes in CLIC3, HSPB1, and glyceraldehyde-3-phosphate dehydrogenase (GAPDH) have been validated by at least one other protein assay (Table 2).

CLIC3 belongs to the chloride intracellular channel family proteins [74] and is involved in various cellular processes, including cell division [84] and membrane fusion [85]. In the healthy placenta, CLIC3 is found in SCT and EVT [65] and is elevated in extracellular vesicles (EV) [86] and protein extracts from PE placentas [43]. It remains unclear whether this increase is due to the preterm nature of PE placentas or is specific to PE pathology.

HSPB1, also known as heat shock protein 27 (HSP27), is a member of the small heat shock protein family and regulates the transcriptional activation of the heat shock response [87]. It is expressed in various tissues and localized in the SCT and stromal cells of the placenta [65]. PE placentas exhibit elevated levels of HSPB1 in protein extracts [48,88] and increased staining intensity in SCT compared to healthy controls [40].

Because HSPB1 is induced by cellular stressors like hypoxia, its increase in the PE placenta may represent a protective response to facilitate protein folding and suppress apoptosis. Besides quantitative changes, HSPB1 in the PE placenta is more likely to be phosphorylated [40] and localized to the cytosol [49]. The functional implications of HSPB1 phosphorylation status and subcellular location in PE pathology remain unclear. Interestingly, plasma levels of HSPB1 are elevated in PE pregnancies [89,90], and Mendelian randomization analyses suggest that plasma HSPB1 could be a potential causal risk factor for PE [91]. Further research is needed to replicate these findings and investigate whether the placenta contributes to the elevated maternal plasma HSPB1 levels observed in PE. 

GAPDH is another protein consistently found at elevated levels in the PE placental proteome. It is best known for its role in glycolysis and as a ubiquitous housekeeping gene expressed in all cell types [74]. In the placenta, GAPDH localizes to the trophoblasts and stromal cells [65]. Increased GAPDH levels in the PE placenta may indicate upregulated glycolysis in response to hypoxia [92].

In contrast, three PE-associated placental proteins identified using proteomic analysis—ACTG1, ANAX6, and ATIC—have not been corroborated with other protein assays. ACTG1, a member of the actin family, often coexists with beta-actin to form the cytoskeleton and facilitate cell motility [74]. ACTG1 is primarily localized in EVT, SCT, fetal endothelium, and stromal cells of placental villi [65]. Although the role of ACTG1 in the placenta remains unclear, studies in *Actg1*-null mice suggest it is not essential for fetal development, as the null embryos are viable and normal at birth [93]. Knockdown experiments in trophoblast HTR-8/SVneo cells suggest that *ACTG1* downregulation may impair EVT migration and invasion [94]. More data are needed to validate the association between ACTG1 and PE and to elucidate underlying mechanisms.

ANAX6, a calcium-dependent membrane protein, is involved in endosomal fusion and exocytosis [74]. In the placenta, ANAX6 is predominantly expressed by the SCT, particularly on basal and apical membranes [65]. Studies in SCT vesicles suggest that ANAX6 regulates chloride channel function [95], which appears to be disrupted in PE [96]. Therefore, the increased ANAX6 in PE placentas might represent a compensatory mechanism to maintain the chloride conductance necessary for cell membrane functions. Further investigations are needed to explore this hypothesis and to understand the role of chloride channel activity in SCT function and PE risk.

ATIC is a ubiquitously expressed enzyme involved in purine synthesis [74]. In the placenta, it is primarily expressed by the major trophoblast populations (SCT, VCT, and EVT) [65]. It is unclear why ATIC is elevated in the PE placenta, but it may indicate heightened nucleic acid synthesis in trophoblasts.

Figure 1 illustrates the cell-specific localization of PE-associated proteins and their alterations in the PE placenta. As high-throughput proteomic studies uncover more PE-associated proteins, it is crucial to compare and validate findings to guide future research efforts. Significant proteins identified by proteomic studies should be validated by other protein assays (e.g., ELISA, western blotting, and IHC) and replicated across different PE populations and subtypes. Mechanistic follow-up is necessary to fully leverage the power of high-throughput protein discovery in understanding PE pathophysiology.

## 4. Genetic Variants Associated with PE

PE is highly heritable (estimated at 55%), with contributions from both maternal (~35%) and fetal genomes (~20%) [97,98]. In recent years, the first large-scale maternal and fetal GWASs of PE have been published [15,16,17,99,100]. The largest maternal PE GWAS, published in 2023 by Honigberg et al. [15], analyzed 20,064 PE cases of predominantly European ancestry. This study identified thirteen PE risk variants linked to genes involved in angiogenesis, immune function, natriuretic peptide signaling, renal glomerular function, and trophoblast development [15]. Notably, two variants are near genes highly enriched in the placenta, *FLT1* and *WNT3A* [65]. WNT3A belongs to the WNT family of signaling proteins critical for cell fate regulation during embryogenesis [74]. *WNT3A* mRNA is expressed in placental smooth muscle cells but not in SCT or EVT [65]. The precise role of placental WNT3A and WNT signaling in PE risk remains unclear.

Further analysis of the maternal *FLT1* variant indicates that this variant is also associated with PE risk in the fetal genome [15], consistent with findings from previous fetal PE GWAS studies [16,17]. Multinomial modeling of genetic effects in maternal and fetal genomes suggests that the increased PE risk is primarily mediated through the fetal *FLT1* allele [16]. Findings from these emerging well-powered fetal PE GWASs align with the established role of the FLT1 pathway in PE pathogenesis [3,18], providing compelling evidence that dysregulation at the *FLT1* locus, likely in the placenta, contributes causally to disease risk. 

Future research is needed to uncover the molecular mechanisms by which risk variants in *FLT1* and *WNT3A* affect placental gene expression and function in relation to PE. The placenta is not part of the Genotype Tissue Expression (GTEx) project, which catalogs genetic loci affecting gene expression (eQTL) across 49 human tissues [101]. Nevertheless, there is a growing number of placental eQTL resources emerging from independent studies [102,103,104,105]. These datasets have been critical in identifying placental genes potentially involved in genotype–phenotype associations related to birth weight [106], placental weight [107], and gestational duration [108]. 

It is worth noting that these placental eQTL datasets vary considerably in sample size, gestational age of placentas, health or disease status, placental cell composition, genome coverage, RNA sequencing depth, and analytical approach. As a result, studies have identified distinct sets of placental eQTLs with limited overlap. Future efforts should focus on validating significant placental eQTLs, standardizing methodologies for eQTL mapping, and extending eQTL investigations to non-European populations and across gestation. Additionally, considering the inherent transcriptomic variation and genetic mosaicism among placental cells [109], generating cell-type-specific placental eQTLs will be crucial for identifying the genes and cell types affected by the maternal and fetal genomes. 

Because many genetic variants can alter protein expression without affecting transcript abundance, it is also essential to identify variants that modify protein quantities (pQTLs) in the placenta. Integrating placental eQTL and pQTL data with GWAS findings will provide crucial insights into the biological relevance of PE variants and the role of the placenta in mediating genetic risk.

## 5. Research Gaps and Future Directions

Over the past two decades, significant progress has been made in unraveling the genetic, cellular, and molecular signatures of PE, underscoring the central role of the placenta in its pathophysiology. Large-scale GWASs involving maternal and fetal genomes have begun to elucidate the genetic basis of PE, identifying multiple common risk variants, including those associated with genes enriched in the placenta (e.g., *FLT1* and *WNT3A*). Emerging data indicate that these common PE risk variants often co-occur within critical biological pathways implicated in PE, such as VEGF signaling and immune function, reinforcing the polygenic nature of PE [110]. Given the detrimental effect of PE on reproductive fitness, the most deleterious variants are likely under negative selection, making them rare in the population. Integrating whole genome sequencing into PE genetic research holds promise for discovering low frequency (0.01–5%) and rare genetic variants (<0.01%) not captured by GWASs that contribute to PE risk.

Given the disproportionate burden of PE in non-European populations, particularly for Black individuals [111], increased efforts are needed to study individuals from diverse racial and ethnic backgrounds to enhance genetic discovery, improve genetics-based prediction, and ultimately achieve equity in genomic research. Future studies are needed to replicate the identified PE risk variants in independent cohorts, extend investigations to non-European populations and clinical PE subtypes, and expand GWAS sample sizes by combining all available cohorts to increase the power for discovery.

Despite rapid discoveries with GWASs, questions remain regarding how genetic variants contribute to PE risk and the role of the placenta in mediating the effect. Colocalizing PE risk loci with placental eQTLs and pQTLs could provide insights into the placental genes and proteins involved in PE pathology.

Furthermore, a unique aspect of pregnancy genetics is the interaction between the maternal and fetal genomes. Multiple lines of evidence point to the role of maternal–fetal genetic incompatibility in PE etiology. Nulliparity and conceptions using donor sperm and/or donor oocytes are known risk factors for PE [112,113,114]. Additionally, maternal immune tolerance of fetal EVT is primarily mediated by interactions between maternal killer cell immunoglobulin-like receptors (KIRs) and fetal human leukocyte antigens (HLAs) [115]. Both *HLA* and *KIR* genes are highly polymorphic, and specific allelic combinations are more frequent in PE pregnancies [115,116]. A detailed understanding of how maternal–fetal genetic incompatibility influences placental immune response and PE risk remains a significant gap in PE genetic research.

Transcriptomic and proteomic studies have identified numerous genes and proteins altered in the PE placenta. Recent scRNA-seq studies have highlighted the critical role of EVT and SCT in PE pathology, revealing changes in placental cell abundances and unique gene signatures in EOPE and LOPE placentas. However, more work is needed to validate these findings, standardize placental cell type annotation, and examine other PE subtypes, including HELLP syndrome.

The adoption of spatial omics in placental research, as demonstrated with studies on EVT development [117], spiral artery remodeling [118], and terminal villi morphology [119], holds promise for mapping the cellular expression profiles to specific locations within the placenta. By preserving the spatial context of gene expression, this approach enables investigation into cell–cell interactions and location-specific gene functions in both normal and pathological pregnancies, including PE.

Multi-omic integration has revealed distinct correlations among the plasma transcriptome, immunome, and microbiome in pregnancy and postpartum [120]. Given the systemic nature of PE, analyzing correlation networks between the placenta and maternal plasma could elucidate coordinated interactions across different organ systems and identify circulating markers of PE progression in the placenta.

Regardless of the approach to discovery, thorough follow-up studies are essential to establish mechanistic links to PE. One major limitation of studies using placental tissue is that its phenotype represents an end-point outcome, precluding the investigation of disease progression [121]. Many animal models, mostly murine, have been developed to investigate potential causal pathways leading to PE [122]. However, PE does not occur spontaneously during gestation in any animals, except humans and apes [123]. Thus, animal models of PE generally utilize surgical and pharmaceutical manipulations to induce PE and frequently focus on a single feature of PE rather than the multi-faceted presentation in humans [122]. Moreover, fundamental differences in placental development and physiology between humans and most animal species severely limit the utility of these animal models for studying PE pathophysiology in the human placenta [123].

Human placental cell lines, such as BeWo and HTR-8/SVneo, are commonly used in vitro models for studying PE, but they display different gene expression profiles and physiological characteristics compared to primary trophoblasts [124]. Organoid technology has addressed this limitation by mimicking the temporal and spatial microenvironment in vivo, significantly improving in vitro modeling of trophoblast function, including EVT differentiation, pathogen response, and drug transport [117,125,126,127,128]. Thus, trophoblast organoids offer a promising alternative to traditional cell lines for studying PE-related processes. Improved EV isolation and characterization methodologies in maternal plasma make it a promising non-invasive model to study placental-origin transcripts and proteins in PE etiology and early detection [34,35,129].

Because PE involves multiple additional cell types beyond trophoblasts, it is critical for model systems to preserve cellular arrangements and cell–cell interactions within the placenta. Multi-cellular cultures, such as placental explants [130,131] and organoids [132], enable cell-type-specific characterization of placental physiology. Furthermore, developing placental models amenable to genetic manipulation will be crucial for understanding how maternal and fetal PE variants, both independently and collectively, impact placental gene expression, function, and development. Lastly, analytical approaches capable of integrating multi-omic data will facilitate the identification of shared features and interactive pathways most relevant to PE, offering holistic insights into the molecular underpinnings of PE and potential therapeutic targets.

## Figures and Tables

**Figure 1 ijms-25-09343-f001:**
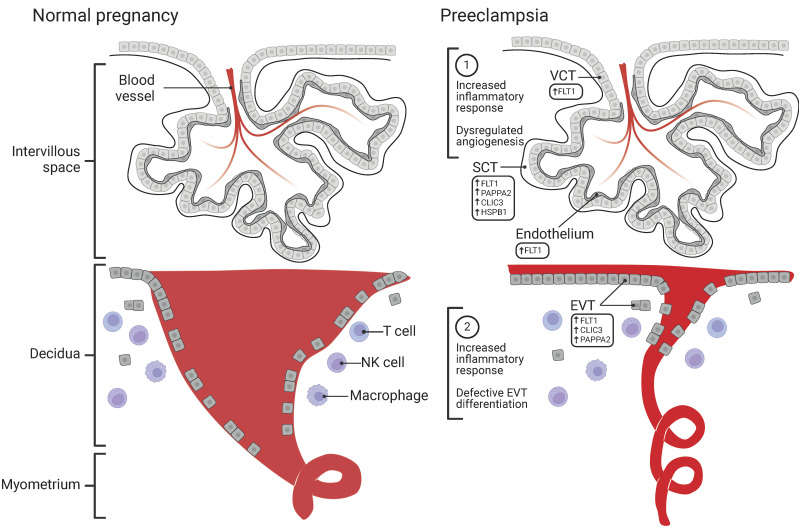
Localization of PE-associated genes, proteins, and pathways in the late gestation placenta. In the fetal villi of the PE placenta, transcriptomic analysis indicates an increased inflammatory response in Hofbauer cells and dysregulated angiogenesis in syncytiotrophoblast (SCT). The signature PE anti-angiogenic protein, fms-like tyrosine kinase 1. (FLT1) is elevated in SCT, villous cytotrophoblasts (VCTs), and endothelial cells. Besides FLT1, SCT also expresses most of the other PE-associated placental proteins identified by proteomic analyses. In the decidua, interstitial extravillous trophoblast (EVT) cells in the stroma and endovascular EVT cells along the vascular lumen collaboratively remove the smooth muscle and endothelium layers of the spiral arteries, transforming them into low-resistance, large-diameter vessels. Interactions between the EVT and uterine-resident immune cells (e.g., natural killer (NK) cells, T cells, and macrophages) are crucial for successful vascular remodeling. In normal pregnancies, spiral artery remodeling extends from the decidua into the myometrium. In PE, this process is incomplete and remains superficial within the decidua. Figure created with Biorender.com.

**Table 1 ijms-25-09343-t001:** scRNA-seq studies of PE placentas.

Year	Study	Sample Description	Sampling Site	Isolation Technique	Number of Single Cells	Sequencing Platform	Cell Types Identified	Major Findings	Data Availability
2017	[28]	PE (28–32 weeks): *n* = 4 Control (38 weeks): *n* = 6	Near cord insertion, fetal membrane removed.	Enzymatic digestion.	Total of 38,497 cells (17,979 from PE, 20,518 from control).	10XGenomics (10XGenomics, Pleasanton CA, USA).	Total of 11 clusters (decidual cell, endothelial cell, vascular smooth muscle cell, stromal cell, DC, macrophage, T cell, erythrocyte, EVT, VCT, SCT).	There is significant heterogeneity in cellular composition in placentas from different individuals, regardless of sampling site or disease status. EVT cells in PE placentas have greater expression and variability of genes in cell death than those in controls.	European Genome-Phenome Archive (accession number: EGAS00001002449).
2021	[27]	PE (34–35 weeks): *n* = 3 Control (38 weeks): *n* = 3	Near cord insertion, fetal membrane removed.	Enzymatic digestion.	Total of 11,518 cells (6434 from PE, 5084 from control).	GEXSCOPE (Singleron, Nanjing, China).	Total of 13 cell clusters (macrophage, VCT, EVT, monocyte, T cell, NK cell, plasma cell, SCT, myofibroblast, fibroblast, erythroblast, proliferating macrophage, endothelial cell).	PE placentas have greater proportion of NK cells and T cells. PE placentas have upregulated immune responses in EVT and VCT cells and increased expression of genes involved in endoplasmic reticulum protein processing in SCT.	Available upon request.
2022	[25]	Severe PE (32 and 36 weeks): *n* = 2 Control (38 and 40 weeks): *n* = 7	Near cord insertion.	Enzymatic digestion.	Total of 29,008 cells (16,379 from PE, 12,629 from control).	10XGenomics (10XGenomics, Pleasanton, CA, USA).	Total of 11 cell types (neutrophil, EVT, VCT, SCT, macrophage, monocyte, two types of epithelial cells, NK cell, erythroid-like cell, fibroblast).	Trophoblast populations in PE have distinct gene and pathway expression. EVT cells in PE exhibit gene signatures in pro-inflammatory, immune and oxidative stress-related pathways. Regulatory network analysis identifies transcription factors and potential target genes in EVT populations.	GEO repository (accession number: GSE173193).
2023	[24]	PE decidua (37 and 38 weeks): *n* = 2 Control decidua (34 and 37 weeks): *n* = 2	Placental decidua sampled from multiple sites from maternal surface.	NA	Total of 101,250 cells (54,776 from PE, 46,474 from control).	BD Rhapsody (BD Life Sciences, San Jose CA, USA).	Total of 22 cell clusters (dendritic cell, decidual stromal cell, endothelial cell, erythrocyte, 4 types of EVT, granulocyte, 2 types of Hofbauer cells, monocyte, macrophage, 3 types of NK cells, NK T cell, cytotoxic T cell, plasma cell, SCT, VCT, fibrocyte).	PE decidua has increased proportions of macrophages, neutrophils and reduced cytotoxic T cells. PE decidua shows functional dysregulation in most cell clusters including reduced ribosomal function in NK cells and abnormal HLA expression. PE decidua shows enhanced ligand-receptor interactions between EVT and macrophages.	National Genomics Data Center (accession number: HRA004699)
2023	[26]	LOPE placenta (37 weeks): *n* = 3 Control placenta (39–40 weeks): *n* = 3 LOPE decidua (35–37 weeks): *n* = 3 Control decidua (38–41 weeks): *n* = 4	Near cord insertion.	Enzymatic digestion.	Total of 12,255 placental cells (5592 from PE, 6663 from control). Total of 20,322 decidual cells (9789 from PE, 10,533 from control).	10XGenomics (10XGenomics, Pleasanton CA, USA).	Total of 12 cell types in the placenta (proliferating VCT, VCT, EVT, SCT, DC, macrophage, T cells, NK cell, B cell, vascular smooth muscle cell, fibroblast, erythroblast). Total of 14 cell types in the decidua (decidual stromal cell, 2 types of fibroblasts, 2 types of T cells, 2 types of NK cells, B cells, antigen presenting cell, lymphatic endothelial cell, erythroblast, granulocyte, endometrial epithelial cell, smooth muscle cell).	PE placentas have downregulated development pathways in trophoblasts, defective pathways involved in EVT invasion, and increased inflammatory response. PE decidua has upregulated epithelial to mesenchymal transition (indicative of defective decidualization), increased inflammatory responses, and reduced immune functions.	GITHUB: https://github.com/JustMoveOnnn/preeclampsia/tree/main/single_cell_matrix/data (accessed on 2 March 2024)
2023	[13]	EOPE (24–35 weeks): *n* = 10 LOPE (37–40 weeks): *n* = 7 EOPE Control (24–34 weeks): *n* = 3 LOPE Control (38–40 weeks): *n* = 6	Decidua removed; placental cotyledon sampled near cord insertion. Placental villi isolated from both maternal and fetal side of cotyledon.	Enzymatic digestion.	Total of 86,752 cells from all placentas (~3337 cells/placenta).	10XGenomics (10XGenomics, Pleasanton CA, USA).	Total of 6 placental cell classes: trophoblast, lymphoid cell, myeloid cell, stromal cell, endothelial cell, mural cell.	Placental cell types and cell-specific gene expression are markedly dysregulated in early but not late PE. Angiogenic imbalance in FLT1/PGF is present only in SCT of EOPE, not in other trophoblast populations or LOPE. Stromal cells in early PE exhibit increased stress response and inflammation while vascular cells display a stress-induced and antiproliferation state. Macrophages of both maternal and fetal origin contribute to pro-inflammation in early PE placentas.	Figshare: https://doi.org/10.6084/m9.figshare.23264102.v1. (accessed on 2 March 2024)

Abbreviations: scRNA-seq, single-cell RNA sequencing; PE, preeclampsia; NA, not available; EVT, extravillous trophoblast; VCT, villous cytotrophoblast; SCT, syncytiotrophoblast; EOPE, early-onset preeclampsia; LOPE, late-onset preeclampsia; NK cell, natural killer cell; DC, dendritic cell; FLT1, fms-like tyrosine kinase 1; PlGF, placental growth factor.

**Table 2 ijms-25-09343-t002:** Placental expression profiles of 11 proteins with consistent patterns of change in proteomic studies of PE and control placentas.

Gene Name	Placental Cells with Transcript Expression *	Placental Cells with Protein Expression *	Fold Change (PE/Control)	Sample Size and Gestational Age of Placental Proteomic Studies	Validation of Proteomic Findings in PE Placentas
*CLIC3*	SCT, EVT, VCT	SCT, EVT	1.55–2.78	PE (*n* = 10; 35–36 weeks); control (*n* = 10; 39 weeks); fold change = 1.63 [38] PE (*n* = 20; 34 ± 3 weeks); control (*n* = 20; 38 ± 3 weeks); fold change = 2.78 [39] PE (*n* = 10; 35–40 weeks); control (*n* = 10; 37–38 weeks); fold change = NA [40] PE (*n* = 5; 35.9 ± 0.8 weeks); control (*n* = 5; 37.1 ± 1.1 weeks); fold change = 1.55 [41] PE (*n* = 8; gestational age unknown); control (*n* = 1; 28–40 weeks); fold change = NA [42]	ELISA [43]
*HBZ*	Mixed immune cells	Fetal blood cells	1.99–4.97	PE (*n* = 10; 35–36 weeks); control (*n* = 10; 39 weeks); fold change = 1.99 [38] PE (*n* = 20; 34 ± 3 weeks); control (*n* = 20; 38 ± 3 weeks); fold change = 4.97 [39] PE (*n* = 20; 28.2–33.4 weeks); control (*n* = 20; 28.1–35.2 weeks); fold change = 2.98 [44]	NA
*GAPDH*	EVT, VCT, fibroblasts	SCT, VCT, EVT, stromal cells	1.53–2.28	PE (*n* = 20; 34 ± 3 weeks); control (*n* = 20; 38 ± 3 weeks); fold change = 1.53 [39] PE (*n* = 6; 34 ± 6 weeks); control (*n* = 6; 40 ± 1 weeks); fold change = 2.28 [45] PE (*n* = 3; 36.1 ± 1.0 weeks); control (*n* = 6; 36.4 ± 1.6 weeks); fold change > 2 [46]	Western blot [47]
*HSPB1*	Smooth muscle cells, SCT	SCT, stromal cells	>1.31	PE (*n* = 10; 35–40 weeks); control (*n* = 10; 37–38 weeks); fold change = NA [40] PE (*n* = 8;gestational age unknown); control (*n* = 1; 28–40 weeks); fold change = NA [42] PE (*n* = 5; 35.6 ± 2.3 weeks); control (*n* = 5; 37.4 ± 1.4 weeks); fold change = NA [48] PE (*n* = 6; 34 ± 6 weeks); control (*n* = 6; 40 ± 1 weeks); fold change = 1.31 [45] PE (*n* = 3; 36.1 ± 1.0 weeks); control (*n* = 6; 36.4 ± 1.6 weeks); fold change > 2 [46]	IHC [40,48,49] Western blot [40,48]
*ANXA6*	Smooth muscle cells, SCT	SCT	1.62–4.4	PE (*n* = 20; 34 ± 3 weeks); control (*n* = 20; 38 ± 3 weeks); fold change = 3.1 [39] PE (*n* = 25; 38.5 ± 1.4 weeks); control (*n* = 25; 39.3 ± 1.2 weeks); fold change = 1.62 [50] PE (*n* = 1, gestational age unknown); control (*n* = 1, gestational age unknown); fold change = 4.4 [51]	NA
*FLT1*	SCT, EVT	SCT, VCT, fetal endothelial cells	1.38–2.82	PE (*n* = 10; 35–36 weeks); control (*n* = 10; 39 weeks); fold change = 2.82 [38] PE (*n* = 25; 38.5 ± 1.4 weeks); control (*n* = 25; 39.3 ± 1.2 weeks); fold change = 1.41 [50] PE/GDM (*n* = 9; 34.0 ± 1.1 weeks); GDM (*n* = 9; 38.5 ± 1 weeks); fold change = 1.38 [52]	ELISA [53,54] Western blot [53,54] IHC [53,55,56,57]
*ATIC*	SCT, VCT, EVT	SCT, VCT, EVT	1.02–1.5	PE (*n* = 10; 35–36 weeks); control (*n* = 10; 39 weeks); fold change = 1.38 [38] PE (*n* = 20; 34 ± 3 weeks); control (*n* = 20; 38 ± 3 weeks); fold change = 1.50 [39] PE (*n* = 20; 32.5 ± 1.8 weeks); control (*n* = 20; 38.3 ± 1.6 weeks); fold change = 1.02 [58]	NA
*PAPPA2*	SCT, EVT	SCT, EVT	1.22–3.89	PE (*n* = 10; 35–36 weeks); control (*n* = 10; 39 weeks); fold change = 1.57 [38] PE (*n* = 20; 34 ± 3 weeks); control (*n* = 20; 38 ± 3 weeks); fold change = 3.89 [39] PE/GDM (*n* = 9; 34.0 ± 1.1 weeks); GDM (*n* = 9; 38.5 ± 1 weeks); fold change = 1.22 [52]	Western blot [59,60,61,62] IHC [59,60,61]
*ACTG1*	all cell types	SCT, fetal endothelial cells, stromal cells, Hofbauer cells, EVT	0.62	PE (*n* = 5; 35.6 ± 2.3 weeks); control (*n* = 5; 37.4 ± 1.4 weeks); fold change = NA [48] PE (*n* = 5; 36.8 ± 2.6 weeks); control (*n* = 5; 39.0 ± 1.2 weeks); fold change = NA [63] PE (*n* = 5; 35.9 ± 0.8 weeks); control (*n* = 5; 37.1 ± 1.1 weeks); fold change = 0.62 [41]	NA
*FGB*	EVT	Fetal endothelial cells	0.61–0.71	PE (*n* = 25; 38.5 ± 1.4 weeks); control (*n* = 25; 39.3 ± 1.2 weeks); fold change = 0.71 [50] PE (*n* = 20; 28.2–33.4 weeks); control (*n* = 20; 28.1–35.2 weeks); fold change = 0.61 [44] PE (*n* = 5; 35.6 ± 2.3 weeks); control (*n* = 5; 37.4 ± 1.4 weeks); fold change = NA [48]	IHC [64]
*ALB*	EVT	SCT, fetal endothelial cells	0.41–0.72	PE (*n* = 10; 35–36 weeks); control (*n* = 10; 39 weeks); fold change = 0.41 [38] PE (*n* = 10; 35–40 weeks); control (*n* = 10; 37–38 weeks); fold change = NA [40] PE (*n* = 20; 34 ± 3 weeks); control (*n* = 20; 38 ± 3 weeks); fold change = 0.61 [39] PE (*n* = 25; 38.5 ± 1.4 weeks); control (*n* = 25; 39.3 ± 1.2 weeks); fold change = 0.72 [50] PE (*n* = 5; 35.6 ± 2.3 weeks); control (*n* = 5; 37.4 ± 1.4 weeks); fold change = NA [48]	NA

Abbreviations: PE, preeclampsia; SCT, syncytiotrophoblast; EVT, extravillous trophoblast; VCT, villous cytotrophoblast; NA, not available; ELISA, enzyme-linked immunosorbent assay; IHC, immunohistochemistry; GDM, gestational diabetes. * scRNA-seq and IHC data from Human Protein Atlas [65].

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
