# Peer review of "Placental Origins of Preeclampsia: Insights from Multi-Omic Studies"

_ijms, 2024, doi:10.3390/ijms25179343_

Round 1

Reviewer 1 Report

Comments and Suggestions for Authors

In this review article, Chang Cao et al. provide a concise summary of novel interesting findings from multi-omic studies for placental proteins related to preeclampsia. Preeclampsia, a major cause of maternal and neonatal mortality during pregnancy, is characteristic with maternal hypertension and renal dysfunction. The content of the article is very comprehensive and useful to update our knowledge of the research field of the pathophysiology of preeclampsia. However, some minor revisions are required in regards to the following points. 

Concerns:

1.     Page 1, line 44: The abbreviation “sEng” seems not necessary.

2.     Page 2, line 146: Considering the consistency with other expressions for molecules (e.g., FGB), the abbreviation “(ALB)” should be inserted after “albumin”.

3.     Page 4, line 236: Please provide the full name of the enzyme GAPDH (i.e., glyceraldehyde-3-phosphate dehydrogenase) before the abbreviation.

4.     Pages 9-11: In some references, journal names are likely not abbreviated or are incorrectly abbreviated.

Please check the references 2, 7, 11, 15, 16, 18, 19, 20, 21, 22, 23, 27, 41, 43, and 45.

2. Nat. Rev. Dis. Primers

7. Clin. Proteomics

11. J. Clin. Invest.

15. Commun. Biol.

16. Front. Bioeng. Biotechnol.

18. Front. Cell Dev. Biol.

19. Front. Immunol.

20. Mol. Cells

21. Front. Immunol.

22. Mol. Genet. Genomics

23. Proc. Natl. Acad. Sci. USA

27. Nat. Commun.

41. Acta Obstet. Gynecol. Scand.

43. Curr. Protoc. Bioinformatics

45. Int. J. Mol. Sci.

5.     Page 12: The reference 64 could not be found in PubMed.  Is this a meeting report?

Author Response

We thank the reviewer for their time and careful reading of our manuscript. Please find below our point-by-point responses to the questions and comments raised.

Comment 1: Page 1, line 44: The abbreviation “sEng” seems not necessary.

Authors’ Response 1: We have removed this abbreviation.

Comment 2: Page 2, line 146: Considering the consistency with other expressions for molecules (e.g., FGB), the abbreviation “(ALB)” should be inserted after “albumin”.

 Authors’ Response 2: There may be some misunderstanding between fibrinogen and its component fibrinogen beta chain. In our manuscript, we consistently refer to fibrinogen by its full name, while the fibrinogen beta chain is abbreviated using its official gene symbol, FGB.

We have added the abbreviation (ALB) for albumin and used it consistently throughout the paper.

Comment 3: Page 4, line 236: Please provide the full name of the enzyme GAPDH (i.e., glyceraldehyde-3-phosphate dehydrogenase) before the abbreviation.

Authors’ Response 3: Thank you for pointing this out. We have added the definition for GAPDH.

Comment 4: Pages 9-11: In some references, journal names are likely not abbreviated or are incorrectly abbreviated.

Please check the references 2, 7, 11, 15, 16, 18, 19, 20, 21, 22, 23, 27, 41, 43, and 45.

  1. Nat. Rev. Dis. Primers
  2. Clin. Proteomics
  3. J. Clin. Invest.
  4. Commun. Biol.
  5. Front. Bioeng. Biotechnol.
  6. Front. Cell Dev. Biol.
  7. Front. Immunol.
  8. Mol. Cells
  9. Front. Immunol.
  10. Mol. Genet. Genomics
  11. Proc. Natl. Acad. Sci. USA
  12. Nat. Commun.
  13. Acta Obstet. Gynecol. Scand.
  14. Curr. Protoc. Bioinformatics
  15. Int. J. Mol. Sci.

 Authors’ Response 4: We have corrected the previously misused journal abbreviations as per the suggestion.

Comment 5: Page 12: The reference 64 could not be found in PubMed.  Is this a meeting report?

Authors’ Response 5: No, this is a preprint paper in the journal “eLife”. We have corrected the reference and added its DOI as shown below.

Awoyemi, T.; Jiang, S.; Bjarkadóttir, B.; Rahbar, M.; Logenthiran, P.; Collett, G.; Zhang, W.; Cribbs, A.; Cerdeira, A. S.; Vatish, M. Identification of Novel Syncytiotrophoblast Membrane Extracellular Vesicles Derived Protein Biomarkers in Early-onset Preeclampsia: A Cross-Sectional Study. eLife, 2023. DOI: 10.7554/elife.88841.2

Reviewer 2 Report

Comments and Suggestions for Authors

This is a review of multi-omic studies of the pathophysiology of pre-eclampsia. Despite it’s global burden, the pathophysiology of pre-eclampsia is poorly understood and molecular studies are critical to drive forward this field. As such the review topic is of importance and the IJMS is a good fit. The review is well written and highlights the challenges in the field.

Comments as follows:

It would be helpful for the authors to further clarify the scope and methodology of this review at the end of the introduction. Pulling together which areas are and are not reviewed in depth (e.g. currently in lines 69/70), and which areas are reviewed elsewhere in recent reviews in a single place would be helpful for clarity. In novel reviewed areas (e.g. scRNA-seq, GWAS), can the authors clarify the strategy for the literature searches to capture the highlighted papers?

In lines 106-110, the authors highlight one of the contributors to variability is use of term placentas as controls, and highlight the study by Admati et al which used gestation-matched controls. Given that the gestation-matched controls had spontaneous preterm birth < 34 weeks, which is likely to have an inflammatory component even without diagnosed chorioamnionitis, I wonder whether the authors can further comment on the challenges of an appropriate control placenta to preterm pre-eclampsia placentas.

As a Clinician Academic, I would also like to see increased emphasis on the clinical relevance of the findings from molecular studies where relevant and in the discussion pathways to translational benefit. Molecular tests are being integrated into clinical practice, for example, PAPP-A and PlGF (inversely related to FLT) are used in the FMF pre-eclampsia prediction algorithm (https://fetalmedicine.org/research/assess/preeclampsia/first-trimester), used in some centres in the UK and internationally to guide aspirin prophylaxis. Furthermore, high quality RCTs have established PlGF or flt/PLGF ratio for the diagnosis of pre-eclampsia, some example references are:

-          https://www.nejm.org/doi/10.1056/NEJMoa1414838?url_ver=Z39.88-2003&rfr_id=ori:rid:crossref.org&rfr_dat=cr_pub%20%200www.ncbi.nlm.nih.gov

-          https://www.thelancet.com/journals/lancet/article/PIIS0140-6736(18)33212-4/fulltext

Finally it would be useful to further highlight the importance of diversity in genomic and other -omic studies. The disproportionate burden of pre-eclampsia in women of Black ethnicity/African ancestry is not discussed and should be added to the introduction. Further details would highlight the scale of the inequity – for example in the 2023 Honingberg GWAS, just 20 pre-eclampsia cases were in women of Black ethnic backgrounds. The scientific community needs to do more to ensure that samples are diverse and representative of the populations with the burden of disease, so that multi-omic prediction and targeted treatment approaches apply to diverse populations rather than the current issue of Eurocentric bias.

Author Response

We thank the reviewer for their time and careful reading of our manuscript. Please find below our point-by-point responses to the questions and comments raised.

Comment 1: It would be helpful for the authors to further clarify the scope and methodology of this review at the end of the introduction. Pulling together which areas are and are not reviewed in depth (e.g. currently in lines 69/70), and which areas are reviewed elsewhere in recent reviews in a single place would be helpful for clarity. In novel reviewed areas (e.g. scRNA-seq, GWAS), can the authors clarify the strategy for the literature searches to capture the highlighted papers?

Authors’ Response 1: We have added the following description and revised the last paragraph in the introduction per the reviewer’s suggestion. 

Page 4 , Lines 25-29:

In particular, transcriptomic studies of bulk placental tissue have identified many PE-associated genes, transcription factors, and metabolic pathways (Benny 2020; Vennou 2020; Vaiman 2013). However, meta-analyses consistently show a lack of consensus among studies (Vennou 2020; Vaiman 2013), and the specific functions of these PE-associated genes and pathways in the placenta remain poorly characterized and under-reviewed.

Benny, P.A., et al., A review of omics approaches to study preeclampsia. Placenta, 2020. 92: p. 17-27.

Vennou, K.E., et al., Meta-analysis of gene expression profiles in preeclampsia. Pregnancy Hypertension, 2020. 19: p. 52-60.

Vaiman, D., R. Calicchio, and F. Miralles, Landscape of Transcriptional Deregulations in the Preeclamptic Placenta.PLOS ONE, 2013. 8(6): p. e65498.

Page 5, Lines 41-48:

We summarize recent findings from scRNA-seq studies of PE placentas, identified through a PubMed search using the keywords “placenta,” “preeclampsia,” and “single-cell RNA sequencing.” We examine the literature on placental expression and function of proteins consistently altered in the PE placental proteome, exploring their mechanistic links to PE pathophysiology. Additionally, we discuss the implications of maternal and fetal GWAS findings for identifying placental cells and genes that confer increased PE risk. Lastly, we outline research gaps and propose how novel analytic approaches and multi-omic data can enhance our understanding of PE.

Comment 2: In lines 106-110, the authors highlight one of the contributors to variability is use of term placentas as controls, and highlight the study by Admati et al which used gestation-matched controls. Given that the gestation-matched controls had spontaneous preterm birth < 34 weeks, which is likely to have an inflammatory component even without diagnosed chorioamnionitis, I wonder whether the authors can further comment on the challenges of an appropriate control placenta to preterm pre-eclampsia placentas.

Authors’ Response 2: Thank you for highlighting the important issue regarding the selection of control placentas for early onset preeclampsia. The study by Admati et al. utilized placentas from spontaneous preterm deliveries that showed no placental pathologies based on placental cultures and excluded cases with maternal chronic disease or fetal malformations. However, we agree with the reviewer that, despite these exclusion criteria, these control placentas may still have heightened inflammation and other undiagnosed abnormalities that are not present in healthy placentas.

We have added the following observation to relevant section:  

Page 7, Lines 97-102:

However, it is important to note that the control placentas for EOPE in the study Admati et al. were from spontaneous preterm deliveries, which may exhibit heightened inflammation and other metabolic abnormalities compared to normal gestation. Therefore, while gestational age-matched placentas help reduce variability, they may not fully capture the transcriptomic differences between PE and healthy placentas. More data is needed to establish the normal expression profiles of placentas in early gestation.

Comment 3: As a Clinician Academic, I would also like to see increased emphasis on the clinical relevance of the findings from molecular studies where relevant and in the discussion pathways to translational benefit. Molecular tests are being integrated into clinical practice, for example, PAPP-A and PlGF (inversely related to FLT) are used in the FMF pre-eclampsia prediction algorithm (https://fetalmedicine.org/research/assess/preeclampsia/first-trimester), used in some centres in the UK and internationally to guide aspirin prophylaxis. Furthermore, high quality RCTs have established PlGF or flt/PLGF ratio for the diagnosis of pre-eclampsia, some example references are:

-          https://www.nejm.org/doi/10.1056/NEJMoa1414838?url_ver=Z39.88-2003&rfr_id=ori:rid:crossref.org&rfr_dat=cr_pub%20%200www.ncbi.nlm.nih.gov

-          https://www.thelancet.com/journals/lancet/article/PIIS0140-6736(18)33212-4/fulltext

Authors’ Response 3: This is a great suggestion. We have added the following text to highlight the clinical application of the placentally derived PE biomarkers. 

Page 4 , Lines 14-23:

The identification of placental-origin PE biomarkers has improved PE prediction in clinical settings. For instance, measuring serum PIGF (Duhig 2019) and the sFlt1/PIGF ratio (Zeisler 2016) enhances the diagnosis and prediction of PE in pregnant individuals being evaluated for PE. Prediction models that incorporate serum PIGF with clinical risk factors, such as the model developed by the Fetal Medicine Foundation in the United Kingdom, have shown high predictive value for PE across racially diverse populations worldwide (Tan 2018; Chaemsaithong 2022). These models are also increasingly adopted by regional and international guidelines to guide PE management strategies, including aspirin prophylaxis (Poon 2019). As many at-risk pregnancies are still missed by current PE prediction models, ongoing work in improving prediction remains critical. 

References:

Duhig KE, Myers J, Seed PT, Sparkes J, Lowe J, Hunter RM, Shennan AH, Chappell LC; PARROT trial group. Placental growth factor testing to assess women with suspected pre-eclampsia: a multicentre, pragmatic, stepped-wedge cluster-randomised controlled trial. Lancet. 2019 May 4;393(10183):1807-1818. doi: 10.1016/S0140-6736(18)33212-4.

Zeisler H, Llurba E, Chantraine F, Vatish M, Staff AC, Sennström M, Olovsson M, Brennecke SP, Stepan H, Allegranza D, Dilba P, Schoedl M, Hund M, Verlohren S. Predictive Value of the sFlt-1:PlGF Ratio in Women with Suspected Preeclampsia. N Engl J Med. 2016 Jan 7;374(1):13-22.

Tan MY, Syngelaki A, Poon LC, Rolnik DL, O'Gorman N, Delgado JL, Akolekar R, Konstantinidou L, Tsavdaridou M, Galeva S, Ajdacka U, Molina FS, Persico N, Jani JC, Plasencia W, Greco E, Papaioannou G, Wright A, Wright D, Nicolaides KH. Screening for pre-eclampsia by maternal factors and biomarkers at 11-13 weeks' gestation. Ultrasound Obstet Gynecol. 2018 Aug;52(2):186-195. doi: 10.1002/uog.19112.

Chaemsaithong P, Sahota DS, Poon LC. First trimester preeclampsia screening and prediction. Am J Obstet Gynecol. 2022 Feb;226(2S):S1071-S1097.e2. doi: 10.1016/j.ajog.2020.07.020.

Poon LC, Shennan A, Hyett JA, Kapur A, Hadar E, Divakar H, McAuliffe F, da Silva Costa F, von Dadelszen P, McIntyre HD, Kihara AB, Di Renzo GC, Romero R, D'Alton M, Berghella V, Nicolaides KH, Hod M. The International Federation of Gynecology and Obstetrics (FIGO) initiative on pre-eclampsia: A pragmatic guide for first-trimester screening and prevention. Int J Gynaecol Obstet. 2019 May;145 Suppl 1(Suppl 1):1-33. doi: 10.1002/ijgo.12802.

Comment 4: Finally it would be useful to further highlight the importance of diversity in genomic and other -omic studies. The disproportionate burden of pre-eclampsia in women of Black ethnicity/African ancestry is not discussed and should be added to the introduction. Further details would highlight the scale of the inequity – for example in the 2023 Honingberg GWAS, just 20 pre-eclampsia cases were in women of Black ethnic backgrounds. The scientific community needs to do more to ensure that samples are diverse and representative of the populations with the burden of disease, so that multi-omic prediction and targeted treatment approaches apply to diverse populations rather than the current issue of Eurocentric bias.

Authors’ Response 4: We fully agree with the reviewer regarding the lack of racial diversity in PE GWAS. The original manuscript highlighted the need to expand PE genetics research to non-European populations (Lines 289 and 318).

In response to the reviewer’s suggestion, we have added the following text to further emphasize the burden of PE in other racial/ethnic groups and urge the research community to address this critical gap:

Page 16, Lines 313-316:

Given the disproportionate burden of PE in non-European populations, particularly Black individuals (Fasanya 2021), increased efforts are needed to study individuals from diverse racial and ethnic backgrounds to enhance genetic discovery, improve genetics-based prediction, and ultimately achieve equity in genomic research.

 Reference:

Fasanya HO, Hsiao CJ, Armstrong-Sylvester KR, Beal SG. A Critical Review on the Use of Race in Understanding Racial Disparities in Preeclampsia. J Appl Lab Med. 2021 Jan 12;6(1):247-256. doi: 10.1093/jalm/jfaa149. PMID: 33227139; PMCID: PMC8516080.
